# Hyaluronic Acid Interactions with Pork Myofibrillar Proteins in Emulsion Gel-Type Systems

**DOI:** 10.3390/molecules30102230

**Published:** 2025-05-20

**Authors:** Marzena Zając, Lei Zhou, Magdalena Mika, Ziyi Yang, Jingyu Wang, Ye Tao, Wangang Zhang

**Affiliations:** 1Department of Animal Product Technology, Faculty of Food Technology, University of Agriculture, 31-120 Kraków, Poland; marzena.zajac@urk.edu.pl; 2State Key Laboratory of Meat Quality Control and Cultured Meat Development, College of Food Science and Technology, Nanjing Agricultural University, No. 1 Weigang, Xuanwu District, Nanjing 210095, China; lei.zhou@hunau.edu.cn (L.Z.); 2023208032@stu.njau.edu.cn (Z.Y.); 2021208028@stu.njau.edu.cn (J.W.); 2021208023@stu.njau.edu.cn (Y.T.); 3School of Food Science and Technology, Hunan Agricultural University, Changsha 410127, China; 4Department of Biotechnology and General Food Technology, Faculty of Food Technology, University of Agriculture, 31-120 Kraków, Poland; magdalena.mika@urk.edu.pl

**Keywords:** myofibrillar proteins, hyaluronic acid, gel, emulsion, texture

## Abstract

Health benefits associated with hyaluronic acid, along with its properties such as water-binding capacity and antimicrobial activity, suggest that incorporating it into meat systems could provide a basis for formulating functional meat products. This study aimed to evaluate the properties of myofibrillar protein gels and emulsions with varying concentrations of hyaluronic acid. The results indicate that increasing the hyaluronic acid concentration (0.008% to 0.04%) does not significantly affect the cooking loss, while a concentration of 0.08% enhances cooking loss. This, in turn, increased gel hardness, while the water-holding capacity remains unaffected. Cryo-scanning electron microscopy (Cryo-SEM) images revealed a partial disruption of the gel structure, with rising hyaluronic concentrations. In pork myofibrillar protein emulsions, smaller droplets and higher stability were observed after HA incorporation. Samples containing hyaluronic acid were more viscous and exhibited shear-thinning properties. Overall, the hyaluronic acid used in this study improved emulsion properties, whereas the gel structure was compromised.

## 1. Introduction

Apart from widespread cosmetic applications of hyaluronic acid (HA) (being the second most used dermal filler after Botox treatment), it is also recommended for oral implementation [1]. HA or its salts are approved as additives in Japan, South Korea, and China, as their safety has been confirmed in numerous trials [2]. The increasing popularity of HA has prompted the investigation of its effects when incorporated into the food matrix, laying the groundwork for the production of functional foods. To propose HA as an industrial additive, it is essential to understand its interactions with food product proteins and its roll in oil-in-water (o/w) emulsions. Despite our preliminary tests in 2017 [3], the gap in research related to meat systems still exists.

Hyaluronic acid is a naturally occurring polysaccharide present in most connective tissues and consists of repeatable units of β4-glucuronic acid and β3-N-acetylglucosamine [1]. In an aqueous solution, HA adopts a secondary structure characterized by a left-handed helix. Due to hydrophobic interactions and intermolecular hydrogen bonding, the HA double helix organizes into a double chain, which then forms a three-stage β-sheet structure. This distinctive configuration allows HA to retain water efficiently, absorbing amounts that exceed its own weight and thereby forming a gel [4]. In the living organism, HA of high molecular weight plays a structural role, as it is able to bind 10 to 10,000 times more water than its weight [5]. HA in solution adopts an ordered structure, where each disaccharide unit twists 180° relative to the adjacent parts of the chain. This unique conformation gives HA remarkable rheological properties, strong hydrophilicity, and the capacity to bind and retain large amounts of water molecules [6]. Thus, HA can serve as a yield-increasing and texture-improving agent in meat processing [7,8,9]. Moreover, considering its recommended supplementation, HA could help to produce functional meat products for a healthier society.

High water-holding capacity, low cooking losses, and the ability to form strong gels are essential for the meat industry. A lot of effort is devoted to applying additives and/or techniques to maintain acceptable production yields and high economic results [10]. Meat batters are considered oil-in-water emulsions [11]. Theoretically, the fat globules are surrounded by myofibrillar proteins, which are denatured during thermal treatment, stabilizing the system [12]. Sometimes, these batters are treated as pseudo-emulsions, as they do not conform to classic emulsion characteristics. According to the other theory, fat is trapped in the protein–water gel system, representing a dispersion of fat particles in a matrix of solubilized proteins and water [13]. Myofibrillar proteins are crucial components of muscle fibers and are considered excellent emulsifying agents. They are important in maintaining, among other traits, meat product tenderness, juiciness, and water-holding capacity. In meat products that do not contain any emulsifiers, they play a key role in maintaining the product’s structure. Therefore, in traditional processing, the lean meat-to-fat ratio must be carefully managed to produce a high-quality product. Nonetheless, incorporating components other than meat may change the stability of such a system [14]. Therefore, it is worth using agents supporting MPs in their role as emulsifiers.

Heating, as a typical procedure in meat processing, causes protein denaturation and water loss. Therefore, clearly, the meat industry is interested in using additives, including polysaccharides, to manage production losses [15]. Polysaccharides may change the arrangement of proteins due to weak interactions, which could be a cause of steric hindrance. The process of emulsification allows the protein to quickly attach to the oil–water interface, which enhances the viscosity of both the continuous phase and the interfacial layer. The complexes of proteins with polysaccharides show different adsorption behaviors and rheological and emulsifying properties compared to pure biopolymers. During the process of emulsification, protein is adsorbed at the oil–water interface, which consequently increases the viscosity of the emulsion [16]. HA, as known from its high water-binding ability, seems to be a good candidate to serve as a yield-increasing additive. HA exhibits hydrophilic properties due to its –COOH and –OH groups [15,17,18]. It possesses a high water solubility and can form viscous solutions. HA can easily form hydrophilic and water-soluble salts with cations such as Na^+^, K^+^, Mg^+^, and Ca^2+^. There are hydrogen bonds between water molecules and acetylamino/carboxyl groups of HA, which stabilize the structure of HA and contribute to the firmness of the polymer system [6].

The basis of the current study was our previous experience with homogenized sausages produced with HA addition. In that study, 10–100 mg of sodium hyaluronate (MW 1.27 MDa) produced by *Streptococcus zooepidemicus* were incorporated into 1 kg of pork meat batter. The addition of HA resulted in an increased cooking loss in all of the cases and decreased texture properties. We decided to investigate the interactions between myofibrillar proteins and HA to determine the possible explanation for this phenomenon. This study was designed to determine the effect of hyaluronic acid on myofibrillar protein gelling and emulsifying properties. To the best of our knowledge, this is the first trial to examine these interactions.

## 2. Results and Discussion

### 2.1. Properties of MP and HA Gels

Cooking loss and water-holding capacity results are presented in Table 1. The results indicate that low HA concentrations (samples ×500 to ×100) did not affect cooking losses of myofibrillar protein gels. However, when the proportion of MP to HA increased (samples ×50 and ×10), the cooking loss was significantly increased. This finding was unexpected, as the ability of HA to decrease cooking losses is almost common knowledge. High-hydration ability was confirmed by Kaufmann et al. [9], followed by experiments with various proteins and polysaccharides [19,20]. In the sample where the HA concentration was 500 times smaller than MP, we could observe a decrease in cooking losses. It is possible that an even lower concentration may positively affect this parameter. As presented by Joshi et al. [21], this effect may depend on the molecular weight of HA. This was also one of our hypotheses, as the HA used in our study had a molecular weight of 7.5 MDa. Notably, the structure of hyaluronic acid chains is not uniform. HA can form a three-dimensional structure at concentrations below 1 μg/mL, which tends to cause it to entangle at higher concentrations, influenced by its molecular weight [22]. It seems that after losing water during heating, the remaining fluids were kept in the gel network. Higher WHC is desirable to maintain a moist mouthfeel, as well as high production yields [23]. In our case, the lack of additional losses can be treated as a positive property. In the research conducted on gel emulsions based on whey protein isolate and HA, WHC increased when HA was incorporated; however, it decreased significantly as the HA concentration increased above 0.1% [24]. This proves that higher HA concentrations are not beneficial for the gel/emulsion properties.

The properties of the gels were analyzed using Cryo-SEM images, which are presented in Figure 1. The Control sample (Figure 1A) shows an irregular and sheet structure with large pores, which are typical for heat-induced myofibrillar protein gels [25,26]. The change in structure can be observed on images with the HA (Figure 1B–F). Low concentrations of HA increased the amount of smaller gaps, which may enhance water–protein interactions. Higher HA concentrations (50:1 and 10:1; Figure 1E and Figure 1F, respectively) induced rupturing of the structure, and the typical sheet of MP cannot be observed anymore. This can be accredited to the molecular weight of the HA used in this study. The system became less rigid and porous with increased HA concentrations. HA, due to the presence of carboxyl groups, may have interacted with MP, creating strong ion bonds. This hypothesis partly complies with the results of TPA (Table 2), which showed higher hardness values, as noted by the significant difference between sample ×50 and the Control. It is possible that HA molecules, too big to align with MP, instead of forming chemical bonds, destroyed the filament structure. A similar observation was made by Shen et al. [26], who analyzed the effect of nanocellulose on cull cow myofibrillar protein gels. With an increase in nanocellulose, the structure was changed from layered to loose and disordered, with a higher amount of pores. The concentration of nanocellulose of 5–10 g/kg increased hardness; however, after a further increase (at 15–20 g/kg), the gel hardness decreased. Large pores and coarse structure were observed by Li et al. [27] in myosin gels, while the pores became smaller and evenly distributed after adding curdlan. Also, in this case, it could be observed that curdlan increased the gel strength, which decreased slightly with increased curdlan concentrations. However, the values were much higher compared to the control. Also, Wang et al. [23] observed an interpenetrating double network with increased hardness, resilience, gumminess, and chewiness when sodium alginate was incorporated into soy protein emulsion gels. Looking closer at the SEM images in our study, it is possible to detect a separate network resembling the one presented by Gamini et al. [28].

Texture properties are measured to determine the sample’s mechanical and sensory texture properties. The instrument records resistance, deformation, or the force required to penetrate or compress the material. The texture measurement of myofibrillar protein gels provides insights into their structural integrity, gelation properties, and functional characteristics. These measurements are crucial in food science, particularly for processed meat products, seafood, and plant-based protein alternatives. It may be affected by pH, protein concentration, heating conditions, and various additives [29,30]. TPA results presented in Table 2 confirm the SEM-observed structure. Sample ×50, which lost a large amount of water, was significantly harder, springier, and less adhesive than the Control. All the other samples were comparable. Sample ×50 was significantly chewier compared to the Control, as well as samples ×10 and ×500. A similar increase in texture parameters, followed by a subsequent decrease in these parameters, was observed by Wang et al. [24]. The same conclusion was made that higher HA concentrations interfered with the gel network. Taking into account only texture parameters, sample ×50 could be considered the most desirable. However, high cooking losses discriminated against this sample from the application standpoint.

### 2.2. Properties of HA-Added Meat Emulsions

Myofibrillar proteins are considered excellent emulsifying agents. In meat products that do not contain any emulsifiers, they play a key role in maintaining the product’s structure. Therefore, in traditional processing, the lean meat-to-fat ratio must be carefully managed to produce a high-quality product. However, modern processing aims for higher yields and lower production costs, which are often achieved through the use of meat additives [15,31]. Polysaccharides may change the arrangement of proteins due to weak interactions, which could be a cause of steric hindrance. The process of emulsification allows the protein to quickly attach to the oil–water interface, which enhances the viscosity of both the continuous phase and the interfacial layer. The complexes of proteins with polysaccharides show different adsorption behaviors and rheological and emulsifying properties compared to pure biopolymers. During the process of emulsification, protein is adsorbed at the oil–water interface, which consequently increases the viscosity of the emulsion [16].

The EAI is associated with the ability of an oil-and-water phase to create an emulsion and illustrates the performance of protein–lipid interactions at the oil–water interface. As shown in Table 3, HA increased the adsorption capacity of MP with increased concentrations. Significant differences were noted when higher amounts were applied (sample ×50). Further HA concentration increase caused a slight decrease in EAI. Again, the sample with a HA-to-MP ratio 50 times lower exhibited the most desirable emulsifying properties.

ESI is related to the dispersed and continuous phases of the emulsion, which shows the stability of the emulsion. In contrast to EAI, we can observe the lack of any effect of HA on the emulsion stability, except for the highest applied concentration, at which the stability was significantly lower. According to McClements [32], certain concentrations of polysaccharides accelerated creaming instability due to a depletion mechanism. The effect of the polysaccharide on the emulsion performance depends on the characteristics of the system.

Another measure of emulsion properties, such as creaming, flocculation, or coalescence, is the size and distribution of oil droplets. According to Stoke’s law, the velocity at which a droplet moves is related to the square of its radius. Reducing the size of the droplets can enhance the emulsion stability. Polysaccharides have been found to adsorb onto oil droplets, additionally stabilizing the emulsions against flocculation and coalescence when used along with proteins [33]. HA addition did not significantly change the droplet size (Figure 2) measured with the laser particle size, except for the sample with the highest HA content, which decreased the droplet sizes. Smaller droplets reflect higher resistance to aggregation and gravitational separation [34]. Therefore, it can be concluded that a high HA concentration was beneficial for the system. The droplet size distribution showed more big droplets in samples ×50 and ×200. However, CLSM images (Figure 3) showed much smaller sizes of droplets in all samples containing HA, among which samples ×50 and ×10 were the most distinct.

Zeta potential analysis, which reflects the stability of the MP–HA complexes (Figure 2), did not show any significant differences among samples. This is in contrast with the results obtained by Wang et al. [20], who analyzed casein–hyaluronic acid emulsion gels. In their experiment, the increased HA concentration decreased Zeta potential values, which indicated an enhanced electrostatic interaction between casein and HA, consequently enabling the formation of soluble complexes. Typically, the decrease in positive charges favors the creation of protein aggregates. A colloidal system with a higher zeta potential (either positive or negative) is usually more stable and less prone to particle aggregation, whereas a lower zeta potential promotes flocculation and coagulation due to Van der Waals forces. Assessing zeta potential is a useful technique for determining colloidal stability, and it is commonly used across various industries, such as ceramics, paints, inks, pharmaceuticals, water treatment, and food and beverage emulsions [35,36]. Usually, the Zeta potential values confirm the droplet size results; however, such a confirmation was not shown in our study. Myofibrillar protein emulsion gels formulated with corn oil at pH 6.0 exhibited Zeta potential values between −40 and −20 mV, depending on the MP concentration [37]. The Zeta potential value of hyaluronic acid was 67mV in a study by Wang et al. [38]. The positive charge of proteins combined with negatively charged HA should exhibit electrostatic repulsive forces between polymers, as it was observed in research on MP and sodium alginate [39].

Particle migration, particle size variation, and particle aggregation are major factors causing emulsion destabilization. The TSI is a unique parameter specific to Turbiscan, created for formulators to easily assess and compare the physical stability of different formulations with a single, consistent, and reproducible value. This tool allows for the quantification of all types of destabilization, ensuring reliability and eliminating user dependency. The defuse reflectance depends on the transport length of the photon giving the kinetic information on the separation process [40,41]. The Turbiscan Stability Index represents the total sum of all backscattering or transmission variations across the entire sample caused by destabilization. A higher value indicates greater instability in the sample. Increased emulsion stability with higher HA concentrations was confirmed by a light-scattering detection method, as shown in Figure 4. A clear and consistent increase in emulsion stability was observed with a rising HA concentration. In the research by Wang et al. [38], WPI (whey protein isolate) emulsions stabilized with HA at low HA concentrations (0.1, 0.2, and 0.4%) exhibited a lack of homogeneity, with excess protein being aggregated and the presence of some larger oil droplets in the continuous phase. However, increased concentrations of HA (0.6, 0.8, 1.0, and 1.2%) led to a more uniform droplet distribution, likely due to strong electrostatic repulsions between HA and WPI.

The increased apparent viscosity values of MP–HA emulsions with the HA concentration are presented in Figure 5. Under a high apparent viscosity, higher shearing forces are required to make the system flow [3]. All the samples showed a decreasing trend of the apparent viscosity with an increasing shear rate. This observation is consistent with non-Newtonian characteristics, which exhibit shear-thinning behavior. Based on the results obtained from the software, both the Cross and Carreau–Yasuda models were suitable to describe the flow behavior of the tested emulsion. Both of the models support a hypothesis that the formation and the rapture of structural linkages in the materials are related to a pseudoplastic flow with asymptotic viscosities at zero and infinite shear rates [42]. Similarly, Ching et al. [43] stated that the Carreau and Cross models are the most appropriate for characterizing and forecasting the flow behavior of oil-filled alginate microgel suspensions. According to Kwiatkowski et al. [44], the Carreau model allows obtaining more accurate values of the settling velocity at higher polymer concentrations compared to the power law model. Wang et al. [20] observed the shear-thinning properties of casein–HA emulsion gels, as well as a higher viscosity of gels with increased HA concentration. As stated by Michaud [45], a solution containing 10 g/L of HA exhibited a viscosity 10 times higher than the solvent, and it increased exponentially above the winding point. Higher viscosity can be treated as beneficial for emulsion stability due to the reduced droplet diffusion speed.

## 3. Materials and Methods

### 3.1. Chemical Reagents and Materials

Hyaluronic acid was obtained from Solarbio company (Beijing, China). Meat (pork *M. longissimus dorsi*) was purchased from a local retailer. NaCl, Na_2_HPO_4_, 2 mmol·L^−1^ MgCl_2_, ethylene glycol-bis(2-aminoethylether)-N,N,N′,N′-tetraacetic acid, and dithiothreitol were purchased from Sinopharm Chemical Reagent Co., Ltd. (Shanghai, China); the dextran standards (Sigma Aldrich, St. Luis, MO, USA) and other chemicals were analytical grade.

### 3.2. Extraction of Meat Protein

Pork myofibrillar proteins (MPs) were extracted as explained by Zhou et al. [46]. Briefly, fresh meat was cut into small pieces (around 0.5 × 0.5 × 0.5 cm^3^). The homogenization of meat was performed using a high-speed tissue masher (DS-1, Specimen Model Factory, Shanghai, China), (3 × 10 s) at 10,000 rpm with 4 volumes of buffer I (0.1 mol·L^−1^ NaCl, 10 mmol·L^−1^ Na_2_HPO_4_, 2 mmol·L^−1^ MgCl_2_, 1 mmol·L^−1^ Ethylene glycol-bis(2-aminoethylether)-N,N,N′,N′-tetraacetic acid and 0.5 mmol·L^−1^ dithiothreitol, pH 7.0, 4 °C). Centrifuging was performed on homogenates (4 °C, 2000× *g* for 20 min, Avanti J-26XP, Beckman Coulter, Brea, CA, USA), and the sediments were collected. The procedure was repeated twice using buffer I. The sediments were resuspended in 4 volumes of buffer II (0.1 mol·L^−1^ NaCl, 1 mmol·L^−1^ NaN_3_, pH 6.0, 4 °C), and the homogenates were filtered with gauze (Medical Equipment Factory of Shanghai Medical Instruments Co., Ltd., Shanghai, China). After centrifuging the homogenates at 2000× *g* under 4 °C for 20 min, the sediments were collected, and the above procedures were repeated twice using buffer II. The sediments after 6 times centrifugation were the extracted proteins. The protein concentrations were determined by the Biuret method with bovine serum albumin as the standard.

### 3.3. The Analysis of HA Molecular Weight (MW)

Molecular weight was determined according to the method developed and validated by Suárez-Hernández et al. [47], using HPLC chromatography (Dionex Ultimate 3000, Thermo Fisher Scientific, Waltham, MA, USA) and dextran standards (Sigma Aldrich, St Luis, USA) and column TSKgelGMPWxl (Sigma Aldrich, St. Luis, USA, 300 cm × 7.8 mm × 13 mm). Isocratic elution was carried out using a phosphate buffer (0.05 M, pH 7.2) containing 0.87% NaCl at a flow rate of 0.4 mL/min at 25 °C. A differential refractometer (Knauer 2300, KNAUER, Berlin, Germany) was used for detection. The average MW was determined as 7.5 MDa.

### 3.4. Preparation of MP and HA Gels

The 40 mg/mL of MP (0.6 M NaCl, 10 mM Na_2_HPO_4_, pH 6) was mixed with HA in the following proportions: 1:0.002 (×500), 1:0.005 (×200), 1:0.01 (×100), 1:0.02 (×50), and 1:0.1 (×10), compared to the Control sample, which did not contain HA. The mixtures were homogenized in the homogenizer (PD500-TP, GreenPrima Instruments Co., Ltd., Shanghai, China) at the speed of 10,000 rpm for 1 min and then mixed at 500 rpm for 12 h.

#### 3.4.1. Determination of Cooking Loss

MP and MP–HA solutions were placed in a water bath, heated at 80 °C for 20 min, and then cooled down to 4 °C. After the heating process, the gels were removed, and the surface water was absorbed. The weight difference before and after heating was measured and referred to as the cooking loss.

#### 3.4.2. Water-Holding Capacity

The water-holding capacity (WHC) was determined with a centrifugation method. The gels were centrifuged (10,000× *g*/4 °C/20 min), the released water was removed, and the tubes with gels were weighed again. WHC was calculated as follows:$$\mathrm{WHC}\;(\%)=(1\;-\;\frac{m_{\mathrm{aqueous}}}{m_{1}\;-\;m_{\mathrm{tube}}})\;\times \;100\%$$
where m_tube_ is the weight of the centrifuge tube, m_1_ is the weight of the gel plus centrifuge tube before centrifugation, and m_aqueous_ is the weight of the released water.

#### 3.4.3. Texture Analysis

The texture properties of gels (cylinder of 1 cm × 1 cm) were determined using a texture meter with a P50 probe (TA. XT. Plus, Stable microsystem, Surrey, UK). The pre-test speed, test speed, and post-test speed were controlled at 5, 1, and 5 mm/s, with a 2 s pause between compression cycles, and the strain was set as 75% with 5 g auto force. Each sample was analyzed 4–6 times.

#### 3.4.4. Cryo-Scanning Electron Microscope (Cryo-SEM)

The ultrastructure of gels was analyzed using a Cryo-SEM (SU 8010, Hitachi Corporation, Tokyo, Japan) at 1000× under a 5 kV accelerating voltage. First, conductive carbon glue was applied on the sample stage, and the samples were stuck on the conductive carbon glue. After that, the sample was put into liquid nitrogen slush for 30 s, and then they were transferred to the sample preparation chamber for sublimation gold plating using a low-temperature freezing preparation transfer system under vacuum. The sample was sublimated at −90 °C for 10 min and then sputtered with gold for 60 s at a current of 10 mA.

### 3.5. Preparation of HA-Added Meat Emulsion

The 10 mg/mL of MP (0.6 M NaCl, 10 mM Na_2_HPO_4_, pH 6) was mixed with HA in the following proportions: 1:0.002 (×500), 1:0.005 (×200), 1:0.01 (×100), 1:0.02 (×50), and 1:0.1 (×10), compared to Control sample, which did not contain HA. The mixtures were homogenized in the homogenizer (PD500-TP, GreenPrima Instruments Co., Ltd., Shanghai, China) at the speed of 10,000 rpm for 1 min and then mixed at 500 rpm for 12 h. Afterwards, the emulsions were formulated by mixing with soybean oil (20%, *v*/*v*) and homogenizing at the speed of 10,000 rpm for 1 min.

#### 3.5.1. Emulsifying Activity and Emulsion Stability Indexes

The determination of the emulsifying activity of MP was obtained with the method as described by Pearce and Kinsella [48], with minor modifications. Briefly, the new and post-10 min of 250 μL aliquots were dispersed in 75 mL of 0.1% SDS (*m*/*v*) solution. The absorbance of solutions was measured at a wavelength of 500 nm. The emulsifying activity index (EAI) and the emulsifying stability index (ESI) were calculated by following Equations (1) and (2):(1)$$\mathrm{EAI}\;(m^{2}/g)=\frac{2\;\times \;2.303\;\times \;300}{C\;\times \;\left(1\;-\;\varphi \right)\;\times {\;10}^{4}}\;\times \;A_{0}$$(2)$$\mathrm{ESI}\;\left(\%\right)=100\;\times \;\frac{A_{10}}{A_{0}}$$
where C denotes the protein concentration (g·mL^−1^) before emulsification, and φ refers to the oil volume fraction (*v*/*v*) in the samples. A_0_ and A_10_ represent the absorbance at the start of the experiment and after 10 min, respectively, and 300 is the dilution factor. All tests were conducted in quadruplicate.

#### 3.5.2. Multiple Light-Scattering Measurement

The storage stability (TSI values) of the HA-added MP emulsion was tested using a vertical scan analyzer (Turbiscan Tower, Formulation, Toulouse, France). Freshly prepared emulsions were transferred into cylindrical glass tubes at 25 °C for 3 h.

#### 3.5.3. Rheological Properties

The rheological properties of MP–HA emulsions were tested using a rotational rheometer (MCR302, Anton Paar, Graz, Austria) with a 50 mm parallel plate. The parameters were monitored as follows: gap of 1 mm and shear rate of 10 to 1000 s^−1^. The Cross (3) and Carreau–Yasuda (4) equations were used (OriginLab, Northampton, MA, USA) to calculate the k and the n values:(3)$$\eta \left(\gamma \right)=\eta_{\infty }+\frac{\eta_{0}-\eta_{\infty }}{1+{\left(k\dot{\gamma }\right)}^{n}}$$
where η(γ) is the viscosity at shear rate γ, η_0_ is the zero shear viscosity, η_∞_ is the shear viscosity, k is the time constant, and n is the flow behavior index.(4)$$\eta \left(\gamma \right)=\eta_{\infty }+\frac{\eta_{0}-\eta_{\infty }}{{\left[1+{\left(k\dot{\gamma }\right)}^{a}\right]}^{\frac{n-1}{a}}}$$
where η(γ) is the viscosity at shear rate γ, η_0_ is the zero shear viscosity, η_∞_ is the shear viscosity, k is the time constant, n is the flow behavior index, and a is the transition control factor.

#### 3.5.4. Droplet Size Measurement

The Malvern 3000 dynamic laser particle size analyzer (Malvern Instruments Ltd., Malvern, Nottinghamshire, UK) was used to determine the droplet size using the method of Hu, Xing et al. [49]. The emulsions were analyzed fresh in triplicate. The refractive index and the adsorption of emulsion particles are 1.436 and 0.001, and the refractive index of phosphate buffer is 1.330. The D10, D50, D90, D(3,2), and D(4,3) values were obtained from this test, as well as the droplet size distribution.

#### 3.5.5. Zeta Potential Measurement

The determination of the Zeta potential was documented by a Zetasizer (Nano-ZS90, Malvern Instruments Ltd., Malvern, Nottinghamshire, UK) after diluting 300× with 10 mM Na_2_HPO_4_ (pH 6.0).

#### 3.5.6. Protein and Oil Distribution Determination Using Confocal Laser Scanning Microscope (CLSM)

The emulsions (1 mL) were stained with 80 μL 0.1% (*m*/*v*) Nile Blue and 0.1% (*m*/*v*) Nile Red mixture for 8 h, and the oil droplets were observed using a CLSM (TCS SP8 X, Leica, Wetzlar, Germany). The excitation wavelengths were 488 nm and 633 nm, respectively. The green color represents oil, and the red color represents protein in this work.

### 3.6. Statistical Analysis

All the experiments were conducted in six independent batches. The results were subjected to one-way analysis of variance using Statistica 13 software (Tibco, Palo Alto, CA, USA) after testing the normality using the Shapiro–Wilk test. The differences between means were tested using the post hoc Duncan test at *p* < 0.05.

## 4. Conclusions

This study examined the effects of the hyaluronic acid (HA) concentration on myofibrillar protein (MP) gels, as well as emulsion gels, focusing on the water-holding capacity (WHC), cooking losses, structure, and emulsifying properties—key factors for the meat industry. Results showed that lower HA concentrations improved WHC and reduced cooking losses, while higher concentrations increased losses and disrupted the gel structure. Moderate HA levels (e.g., ×50) enhanced texture but led to excessive cooking loss, limiting practical use. HA improved MP emulsifying properties, increasing the emulsion activity index (EAI) and stability. However, at very high concentrations, steric hindrance reduced emulsion stability (ESI). Light-scattering and rheology confirmed enhanced viscosity and shear-thinning behavior, critical for food formulations. In summary, HA enhances MP gel and emulsion properties, but its concentration must be optimized for balance. Future studies should explore molecular interactions at varying pH and ionic strengths to refine its application in meat processing. Referring to our previous experiment with homogenized sausages, the results indicate that myofibrillar protein interactions with hyaluronic acid are partly responsible for cooking loss and deteriorated texture. These effects strongly depend on the HA concentration and its molecular weight.

## Figures and Tables

**Figure 1 molecules-30-02230-f001:**
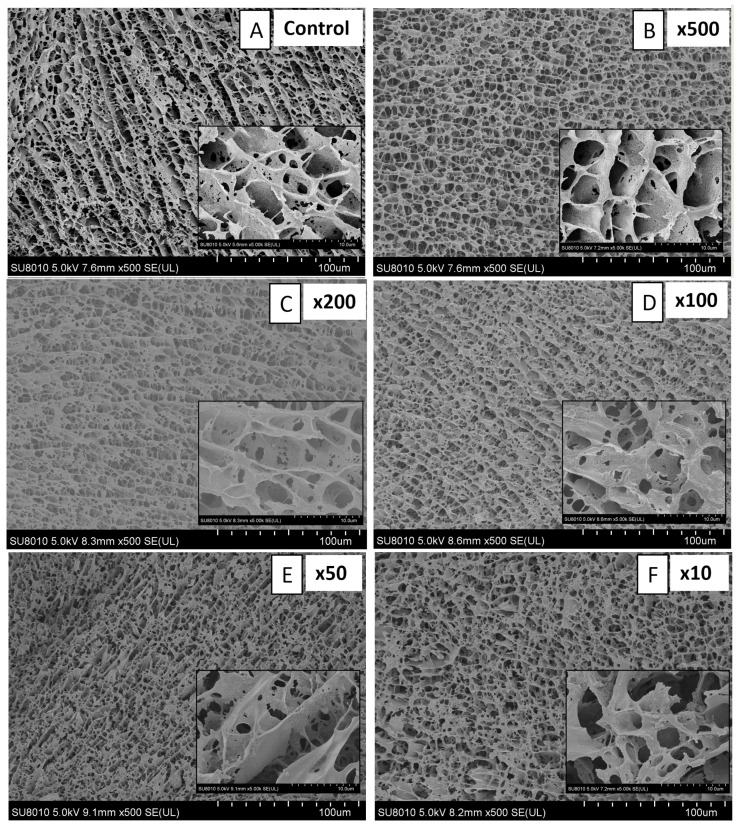
SEM images (magnification, 500× and 5000×) of heat-induced gels with containing myofibrillar proteins and hyaluronic acid in the following proportions: 1:0.002 (×500) (**B**), 1:0.005 (×200) (**C**), 1:0.01 (×100) (**D**), 1:0.02 (×50) (**E**), 1:0.1 (×10) (**F**), and Control sample (**A**).

**Figure 2 molecules-30-02230-f002:**
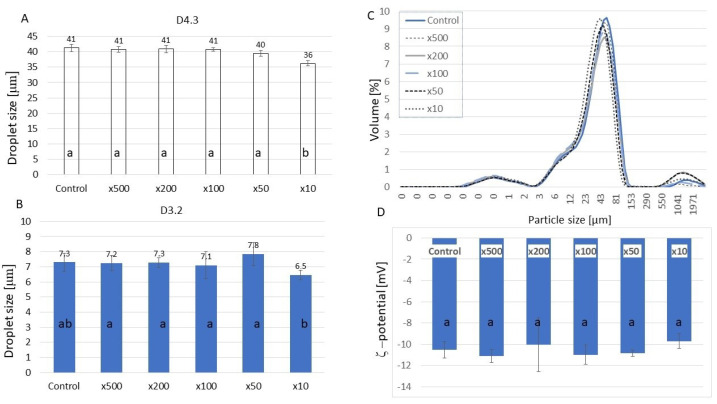
Droplet size (**A**—D4.3; **B**—D3.2), volume distribution (**C**), and ζ-potential (**D**) of myofibrillar emulsions with a variable concentration of hyaluronic acid (mean values ± standard error). ^a.b^—different letters indicate significant differences between means.

**Figure 3 molecules-30-02230-f003:**
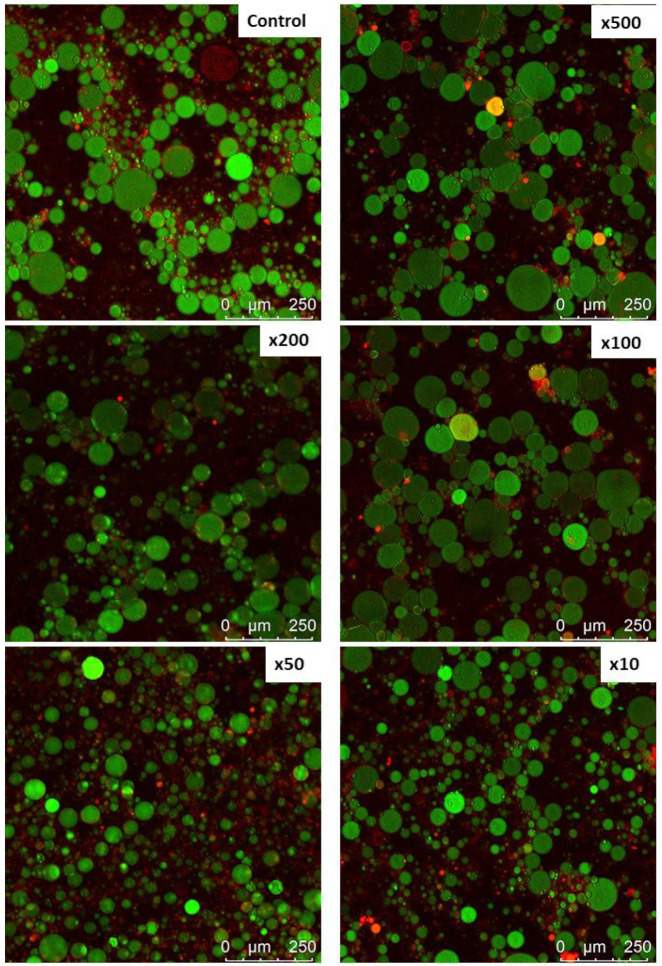
Confocal laser scanning microscopy (CLSM) micrographs of myofibrillar emulsion gels with increasing hyaluronic acid concentrations.

**Figure 4 molecules-30-02230-f004:**
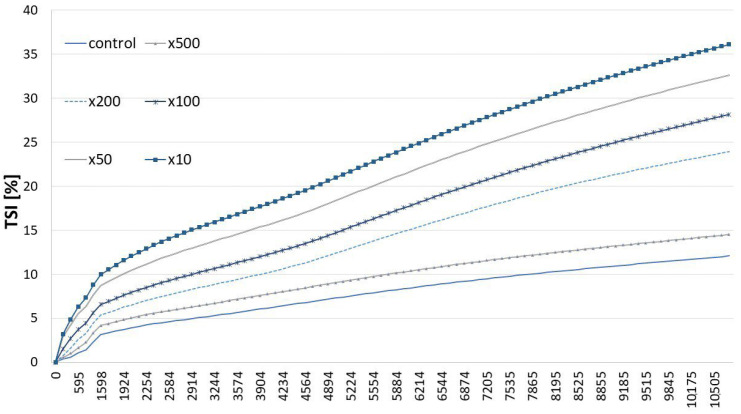
The TSI of myofibrillar emulsions with variable concentration of hyaluronic acid.

**Figure 5 molecules-30-02230-f005:**
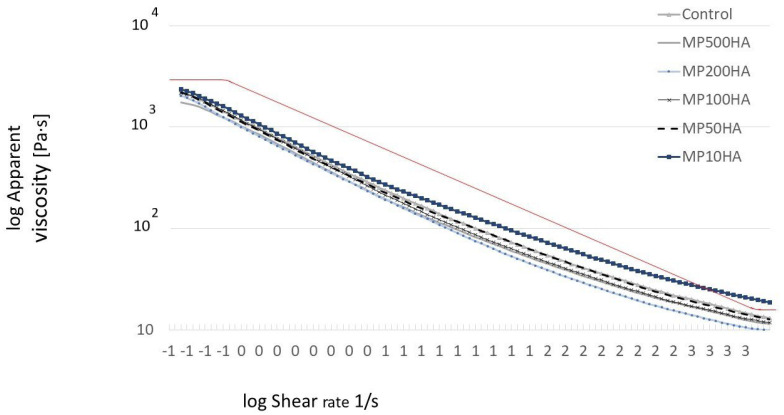
Flow curves of myofibrillar emulsions with variable concentration of hyaluronic acid. Red solid line indicates the Cross model fitting curve.

**Table 1 molecules-30-02230-t001:** Cooking loss and WHC of samples with HA (mean values ± standard errors).

	Cooking Loss [%]	WHC [%] [ns]
Control	6.72 ^b^ ± 0.38	67.14 ± 1.56
×500	6.14 ^b^ ± 0.44	71.12 ± 2.26
×200	5.96 ^b^ ± 0.37	70.27 ± 2.56
×100	9.66 ^b^ ± 1.26	70.40 ± 2.30
×50	22.45 ^a^ ± 1.96	74.71 ± 3.84
×10	35.91 ^a^ ± 2.92	69.05 ± 1.38

^a,b^—different letters in the same column indicate significant differences between means; [ns]—the differences between means are not significant.

**Table 2 molecules-30-02230-t002:** Texture profile of samples with variable amounts of HA (mean values ± standard errors).

Sample ID	Hardness [N]	Adhesiveness	Springiness	Chewiness [N]
Control	1.37 ^b^ ± 0.09	−14.02 ^b^ ± 1.86	0.42 ^b^ ± 0.01	0.21 ^b^ ± 0.03
×500	1.69 ^ab^ ± 0.25	−11.02 ^ab^ ± 0.66	0.50 ^ab^ ± 0.05	0.31 ^bc^ ± 0.09
×200	1.81 ^ab^ ± 0.30	−10.67 ^ab^ ± 0.54	0.52 ^ab^ ± 0.04	0.38 ^ab^ ± 0.11
×100	1.67 ^ab^ ± 0.24	−11.08 ^ab^ ± 0.18	0.52 ^ab^ ± 0.04	0.33 ^abc^ ± 0.08
×50	2.32 ^a^ ± 0.35	−8.24 ^a^ ± 0.51	0.57 ^a^ ± 0.03	0.56 ^a^ ± 0.13
×10	1.75 ^ab^ ± 0.09	−10.63 ^ab^ ± 0.71	0.46 ^ab^ ± 0.02	0.31 ^bc^ ± 0.04

^a,b,c^—different letters in the same column indicate significant differences between means.

**Table 3 molecules-30-02230-t003:** Emulsion activity and emulsion stability indices of samples with variable amounts of HA (mean values ± standard errors).

Sample ID	EAI [g/m^2^]	ESI [%]
Control	1.8 ^c^ ± 0.1	106.8 ^b^ ± 1.5
×500	1.8 ^c^ ± 0.1	105.5 ^b^ ± 2.2
×200	2.0 ^bc^ ± 0.1	103.9 ^b^ ± 2.8
×100	2.1 ^ab^ ± 0.1	111.7 ^b^ ± 3.5
×50	2.2 ^a^ ± 0.1	109.9 ^b^ ± 2.8
×10	2.1 ^ab^ ± 0.0	94.6 ^a^ ± 4.2

^a,b,c^—different letters in the same column indicate significant differences between means.

## Data Availability

The original contributions presented in this study are included in the article. Further inquiries can be directed to the corresponding author.

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
