# Peer review of "Hyaluronic Acid Interactions with Pork Myofibrillar Proteins in Emulsion Gel-Type Systems"

_molecules, 2025, doi:10.3390/molecules30102230_

Round 1
Reviewer 1 Report
Comments and Suggestions for Authors
In the manuscript the authors have described the process of preparation of the myofibrillar proteins -based (MP) functional gels and emulsions with added hyaluronic acid (HA) for oral implementation. Several experimental techniques were used to study the structure of the prepared gels and the stability and the droplet size of the emulsion. Before the publication, I suggest major revision of the manuscript, since the following points are to be addressed:
1) The current work seems to be the routine one and very similar to one, that was previously published by one of the co-authors (Journal of the Science of Food and Agriculture 2017, 97, (8), 2316-2326). The aim of the current manuscript was suggested to be the investigation of the interactions between MA and HA and explanation for the cooking loss and formation the defects in MP-based gels. Nevertheless, I have not found it in the text.
2) Tables 2 and 3 are quite massy. The authors have to demonstrate only the data, that are discussed in the text. Furthermore, some expressions in the table contain too many digits after the dot, e.g. 106.83 ± 1.46. Furthermore, the expression 0.08 ± 0.00 for the resilience has no sense.
3) As the viscosity of the non-Newtonian liquids is defined from the plateau value of the corresponding flow curves, in Figure 5 to apply the fitting with Cross model, the authors have to show the obtained curves in the double logarithmic scale. The recently published paper is suggested to be cited in this case (doi: 10.1063/5.0236308).
4) The quality of the Figures 2,4 and 5 has to be improved to make them more representative.
5) The references in the text have to be made in the same style.
Author Response
Dear Reviewer,
We have prepared a revised version of the manuscript. Based on your suggestions, we have addressed various comments, including major changes to the title, statistical analysis, and sample descriptions. We hope these improvements enhance the manuscript, making it suitable for publication in your journal. All the changes are indicated in red font. Thank you for your time and consideration.
Here is our point-by-point response:
In the manuscript the authors have described the process of preparation of the myofibrillar proteins -based (MP) functional gels and emulsions with added hyaluronic acid (HA) for oral implementation. Several experimental techniques were used to study the structure of the prepared gels and the stability and the droplet size of the emulsion. Before the publication, I suggest major revision of the manuscript, since the following points are to be addressed:
- The current work seems to be the routine one and very similar to one, that was previously published by one of the co-authors (Journal of the Science of Food and Agriculture 2017, 97, (8), 2316-2326). The aim of the current manuscript was suggested to be the investigation of the interactions between MA and HA and explanation for the cooking loss and formation the defects in MP-based gels. Nevertheless, I have not found it in the text.
Response: Thank you for that comment. Indeed, after our first trial with hyaluronan inclusion to a homogenized sausage, we decided to go back to basic analysis and test the interactions between myofibrillar proteins and hyaluronan. Our initial approach was more technological, where the whole meat matrix was used, and the present one is more to investigate pure myofibrillar protein and hyaluronan interactions. It is true that the experimental design is similar but in our opinion, it goes deeper into investigating protein gel and protein emulsion structure (confocal microcopy and cryo-SEM ). The aim was to find our if the results we obtained previously are caused by myofibrillar interactions or other proteins such as e.g. collagen naturally present in meat
- Tables 2 and 3 are quite massy. The authors have to demonstrate only the data, that are discussed in the text. Furthermore, some expressions in the table contain too many digits after the dot, e.g. 106.83 ± 1.46. Furthermore, the expression 0.08 ± 0.00 for the resilience has no sense.
Response: Thank you for this comment. We removed the data with non-significant differences including resilience. We also rounded the numbers.
- As the viscosity of the non-Newtonian liquids is defined from the plateau value of the corresponding flow curves, in Figure 5 to apply the fitting with Cross model, the authors have to show the obtained curves in the double logarithmic scale. The recently published paper is suggested to be cited in this case (doi: 10.1063/5.0236308).
Response: Thank you for this comment. Indeed, presenting the results as log=log charts allowed to see the model alignment much better as well as differences among the tested samples. We modified the figure and added more information in the Results and discussion part. Unfortunately we were not able to get a full access to the suggested paper.
- The quality of the Figures 2,4 and 5 has to be improved to make them more representative.
Response: We increased the fonts and size of chart elements. Hopefully, it improved the image.
5) The references in the text have to be made in the same style.
Response: Thank you for this comment. We corrected the references
Reviewer 2 Report
Comments and Suggestions for Authors
1. Lines81-82, Some references are needed to this effects of two functional groups (-COOH and -OH).
Authors must describe in more detail the experimental method, the purpose of the measurement, and the principle of the measurement. Specifically, the following four points must be mentioned. Other information must also be described in a way that is easy for readers' understanding.
2. Line 246, in Materials and Method, there are no explain general chemical reagents.
3. Line 267, The authors have to explaine why the molecular weight of HA molecule was determine by a liquid chromatography.
4. Line 287, The authors must give the explanation what exactly you are measuring with texture analysis in the manuscript. What does this instrument detect?
5. Line 321, The authors must provide the explanation in the manuscript as to why this measurement can be used to quantify emulsion stability for readers' understanding.
Author Response
Dear Reviewer,
We have prepared a revised version of the manuscript. Based on your suggestions, we have addressed various comments, including major changes to the title, statistical analysis, and sample descriptions. We hope these improvements enhance the manuscript, making it suitable for publication in your journal. All the changes are indicated in red font. Thank you for your time and consideration.
Here is our point-by-point response:
Reviewer 2
- lines81-82, Some references are needed to this effects of two functional groups (-COOH and -OH).
Response: Thank you for this comment. Appropriate references were added.
Authors must describe in more detail the experimental method, the purpose of the measurement, and the principle of the measurement. Specifically, the following four points must be mentioned. Other information must also be described in a way that is easy for readers' understanding.
- Line 246, in Materials and Method, there are no explain general chemical reagents.
Response: This information was added
- Line 267, The authors have to explain why the molecular weight of HA molecule was determine by a liquid chromatography.
Response: High-performance size-exclusion chromatography (HPLC-SEC) is one of the most commonly used methods for determining the molecular weight of polymers and biopolymers, such as hyaluronic acid. This technique enables the separation of molecules based on their hydrodynamic size, which is crucial for analyzing heterogeneous samples with varying degrees of polymerization. Additionally, liquid chromatography allows for the determination of molecular weight distribution, which is significant in the context of the physicochemical and biological properties of hyaluronic acid. The choice of this method can also be justified by its high repeatability, sensitivity, and the possibility of using refractive index detection.
- Line 287, The authors must give the explanation what exactly you are measuring with texture analysis in the manuscript. What does this instrument detect?
Response: We added this information in the Results and Discussion section
- Line 321, The authors must provide the explanation in the manuscript as to why this measurement can be used to quantify emulsion stability for readers' understanding.
Response: We added this information in the Results and Discussion section
Reviewer 3 Report
Comments and Suggestions for Authors
In the current study myofibrillar protein gels and emulsions were analyzed. This study was designed to determine the effect of hyaluronic acid on myofibrillar protein
gelling and emulsifying properties and is of interest. However, the authors should improve results& discussion and conclusion section.
Information from lines 73-86 should be moved from results to introduction.
In figure 1 add a, b, c, and so on for each image to make easier the linkage with the text and the description of the figure can be updated.
Information from lines 153-164 should be moved from results to introduction.
is figure 3 presenting gels or emulsions? it is unclear from its description. Results should be more extensively discussed.
in figure 4 legend should not be presented in the position where a title is usually present, rather move it in right or down.Improve color of axes and legend.
in Chapter 3. Materials and Method- describe materials
include name and producers of devices used for homogenisation in 3.1 and 3.2
in line 277 improve writing.
In 3.3.2 include name of producer and location for centrifuge
in line 283 ,,the tubes with gels were,,?
in 3,3,3 you should mention that the sample was penetrated twice and if there was a pause between penetrations for structure recovery?
Author Response
Dear Reviewer,
We have prepared a revised version of the manuscript. Based on your suggestions, we have addressed various comments, including major changes to the title, statistical analysis, and sample descriptions. We hope these improvements enhance the manuscript, making it suitable for publication in your journal. All the changes are indicated in red font. Thank you for your time and consideration.
Here is our point-by-point response:
In the current study myofibrillar protein gels and emulsions were analyzed. This study was designed to determine the effect of hyaluronic acid on myofibrillar protein gelling and emulsifying properties and is of interest. However, the authors should improve results& discussion and conclusion section.
Information from lines 73-86 should be moved from results to introduction.
Response: Thank you for this comment. This section was moved to Introduction
In figure 1 add a, b, c, and so on for each image to make easier the linkage with the text and the description of the figure can be updated.
Response: Letters were added and the information was linked to the text.
Information from lines 153-164 should be moved from results to introduction.
Response: Thank you for this comment. This section was moved to Introduction
is figure 3 presenting gels or emulsions? it is unclear from its description. Results should be more extensively discussed.
Response: Thank you for this comment. The figure represents emulsions, we did not notice this mistake
in figure 4 legend should not be presented in the position where a title is usually present, rather move it in right or down. Improve color of axes and legend.
Response: We increased the fonts, changed the colour and legend position. Hopefully the figure is improved now.
in Chapter 3. Materials and Method- describe materials
Response: This information was added.
include name and producers of devices used for homogenisation in 3.1 and 3.2
Response: This information was provided
in line 277 improve writing.
Response: The sentence was paraphrased
In 3.3.2 include name of producer and location for centrifuge
Response: This information was provided
in line 283 ,,the tubes with gels were,,?
Response: Of course. Thank you for this comment. It was corrected.
in 3,3,3 you should mention that the sample was penetrated twice and if there was a pause between penetrations for structure recovery?
Response: There was a pause after the first penetration. We added this information
Round 2
Reviewer 1 Report
Comments and Suggestions for Authors
After inspection of the authors' response to my comments I am still not fully satisfied with the manuscript. I would make the following suggestions to improve the text:
1) As the aim of the manuscript was to find out, if the previously obtained results are caused by myofibrillar interactions and not by the other proteins the authors have to underline it directly in the main text;
2) In Figure 5 the experimentally obtained data and corresponding approximations with Cross and Carreu-Yasuda models are to be made with dots and lines, respectively. The authors should choose the right scale to estimate the plateau values of viscosity from the graphs.
Author Response
Dear Reviewer,
Thank you for the additional comments. We have prepared a revised version of the manuscript. We addressed your first comment by adding more information in the conclusions part (indicated in red) reffering to our previous work. We also corrected the rhaology chart adding a fitting curve according to your comment. We hope this version of the manuscript is acceptable in its present form. We highly appretiate your time and help.
Kind regards
Reviewer 2 Report
Comments and Suggestions for Authors
The reviewer thinks that the authors have addressed the reviewers' comments and have corrected the manuscript.
The present form of the revised manuscript should be recommended to accept for publication in Molecules.
Author Response
Dear Reviewer,
Thank you very much for your time and effort in revising this manuscript.
Kind regards